# Effects of multi-functional additives during foam extrusion of wheat gluten materials
Mercedes A. Bettelli [1], Qisong Hu[1], Antonio J. Capezza [1], Eva Johansson [2], Richard T. Olsson[1] & Mikael S. Hedenqvist [1] ✉

To broaden the range in structures and properties, and therefore the applicability of sustainable foams based on wheat gluten expanded with ammonium-bicarbonate, we show here how three naturally ocurring multifunctional additives affect their properties. Citric acid yields foams with the lowest density (porosity of ~50%) with mainly closed cells. Gallic acid acts as a radical scavenger, yielding the least crosslinked/ aggregated foam. The use of a low amount of this acid yields foams with the highest uptake of the body-fluid model substance (saline, ~130% after 24 hours). However, foams with genipin show a large and rapid capillary uptake (50% in one second), due to their high content of open cells. The most dense and stiff foam is obtained with one weight percent genipin, which is also the most crosslinked. Overall, the foams show a high energy loss-rate under cyclic compression (84-92% at 50% strain), indicating promising cushioning behaviour. They also show a low compression set, indicating promising sealability. Overall, the work here provides a step towards using protein biofoams as a sustainable alternative to fossil-based plastic/rubber foams in applications where absorbent and/ or mechanical properties play a key role.

There is a need to replace fossil-based polymers in foams with bio-based alternatives for environmental concerns. Renewable alternatives are therefore actively being researched to replace both thermoplastic foams (e.g., polyethylene and polystyrene) and crosslinked foams (e.g., nitrile rubber and polyurethane). Wheat gluten (WG) obtained as an industrial co/by-product from e.g., bioethanol fuel and wheat starch production is a possible bio-based alternative available at a relatively low cost[1,2]. The good foaming properties of WG are well-known, and commonly observed in bread making, leaving a porous internal structure in bread[3–5]. Its ability to aggregate, crosslink, and polymerize through disulphide bonds yields a quite cohesive material that can be processed with conventional plastic production methods, including extrusion, injection molding, and thermoforming/ compression molding, etc[6–15]. It has also been shown that it is possible to extrude foams based on WG[7,9,10,15–17]. Small and mainly closed cells are obtained from the natural content of moisture in the material; whereas both open and closed larger-cell foams are obtained by the use of a foaming agent, such as ammonium (ABC) or sodium (SBC) bicarbonate[18]. In the previous work it has been shown that extrusion foaming with ABC yields more uniform and porous WG materials than with SBC due to the formation of

both ammonia and carbon dioxide at lower extrusion temperature[18]. It has also been shown on Wheat Gluten and starch-based materials that quite different foam structures can be obtained with variations in material composition and extrusion processing conditions[7,17]. It is also possible to improve mechanical strength, rigidity, cohesion, and water resistance of WG materials by the use of crosslinking agents[19,20].

In the present article, we build on previous work on foam-extruded WG (glycerol-plasticised (G) and foamed with ammonium bicarbonate (ABC)) and explore three non-toxic multi-functional additives to determine if they can contribute to improved overall properties of foam-extruded protein materials, such as mechanical (cushioning/sealing) and liquid absorption properties. The aim was also to identify possible applications for these foams, based on their properties. The three additives are genipin (GNP), gallic acid (GA) and citric acid (CA). Genipin can be used to graft polymers to increase their polarity. It can also work as a crosslinker and possibly also as a plasticiser. Gallic acid can work as a crosslinker, radical scavenger and possibly also as a plasticiser. Citric acid is commonly used to graft/crosslink polymers, but can also act as a plasticiser. It was unknown beforehand, which of these properties would dominate in the WG foams

[1]Department of Fibre and Polymer Technology, Polymeric Materials Division, School of Engineering Sciences in Chemistry, Biotechnology and Health, KTH Royal Institute of Technology, Stockholm, 10044, Sweden. [2]Department of Plant Breeding, The Swedish University of Agricultural Sciences, Box 190, SE-234 22 Lomma, Sweden. ✉e-mail: mikaelhe@kth.se

(considering also any interactions with glycerol and ABC), and the purpose was to determine this. Genipin is a non-toxic natural compound that can be extracted from the fruit *Gardenia jasminoides Ellis* and be used as an alternative to toxic fossil-based dialdehyde crosslinkers, such as glutaraldehyde and formaldehyde[20–25]. Gallic acid (GA) is a phenolic compound recognized for its antioxidant, antibacterial, and anticarcinogenic properties[19,26–28]. It is the main structural component of many dietary polyphenols found in tea, nuts, and grapes[29]. The crosslinking/polymerization ability of gallic acid in proteins have been reported, but studies also shows that[30,31] gallic acid, as mentioned above, can act as a radical scavenger, to delay or inhibit e.g., reactions involving peroxyl radicals, which can be useful to potentially delay/avoid any radical-induced degradation reactions during extrusion[32]. Citric acid has been used to crosslink proteins and polysaccharides, such as starch, chitosan, and cellulose[33,34], as well as proteins such as soy, fish, and collagen[35–37]. It is a bio-based and non-toxic cheap polycarboxylic acid present in citric fruits, such as in lemon and lime[34,38]. Citric acid-induced plasticisation has been reported when present at higher concentration[39].

## Results and discussion
### Foam structure and visual appearance

Figure 1 shows that the foams differed significantly in colour, shape, and size. The extrudates with gallic acid (WG/G/ABC/GA) and genipin (WG/G/ABC/GNP) were dark brown, while the samples produced with citric acid

(WG/G/ABC/CA) and the reference foams (WG/G and WG/G/ABC), had a similar light brown/beige colour to that of the wheat gluten powder. GA exhibits a series of colours, from orange to green, depending on pH, time in the solution under ambient conditions and OH⁻ dissociation and radical formation processes[40]. In reactions between GA and proteins, the colour change originates from enzymatic browning reactions between GA quinone and amino acid residues in the protein, such as cysteine, lysine, methionine, and tryptophan. These reactions consume oxygen and the enzymes' oxidation catalyzes them[41,42]. The highly reactive quinone normally reacts also with other quinones to form brown-coloured species. The addition of genipin resulted in a colour change during the extrusion process, from light yellow to dark brown, indicating a reaction between genipin and WG amino acid residues in the presence of oxygen[20,43]. A colour gradient was clearly observed when viewing the cross-section of WG/G, WG/G/ABC, WG/G/ABC/5GA and WG/G/ABC/5CA (Supplementary Fig. S1). It was also observed that, regardless of the additive used (GA, CA, GNP), the expansion ratio (ER) after the drying of the samples was similar or smaller than that of the samples without the additive (Table 1). The ER in the presence of 1 and 5 wt% GNP was 1.1 and 0.9, and the corresponding ER-values for GA were both 1.2. With the addition of 1 and 5 wt% CA, the ER became 1.5 and 1.6. This should be compared to the ERs of the reference WG/G and WG/G/ABC foams, which were both 1.5. There are several possible reasons for the low expansion ratio; crosslinking in the extruder could "lock" the molecular structure and lubrication/plasticisation could reduce the die swell by

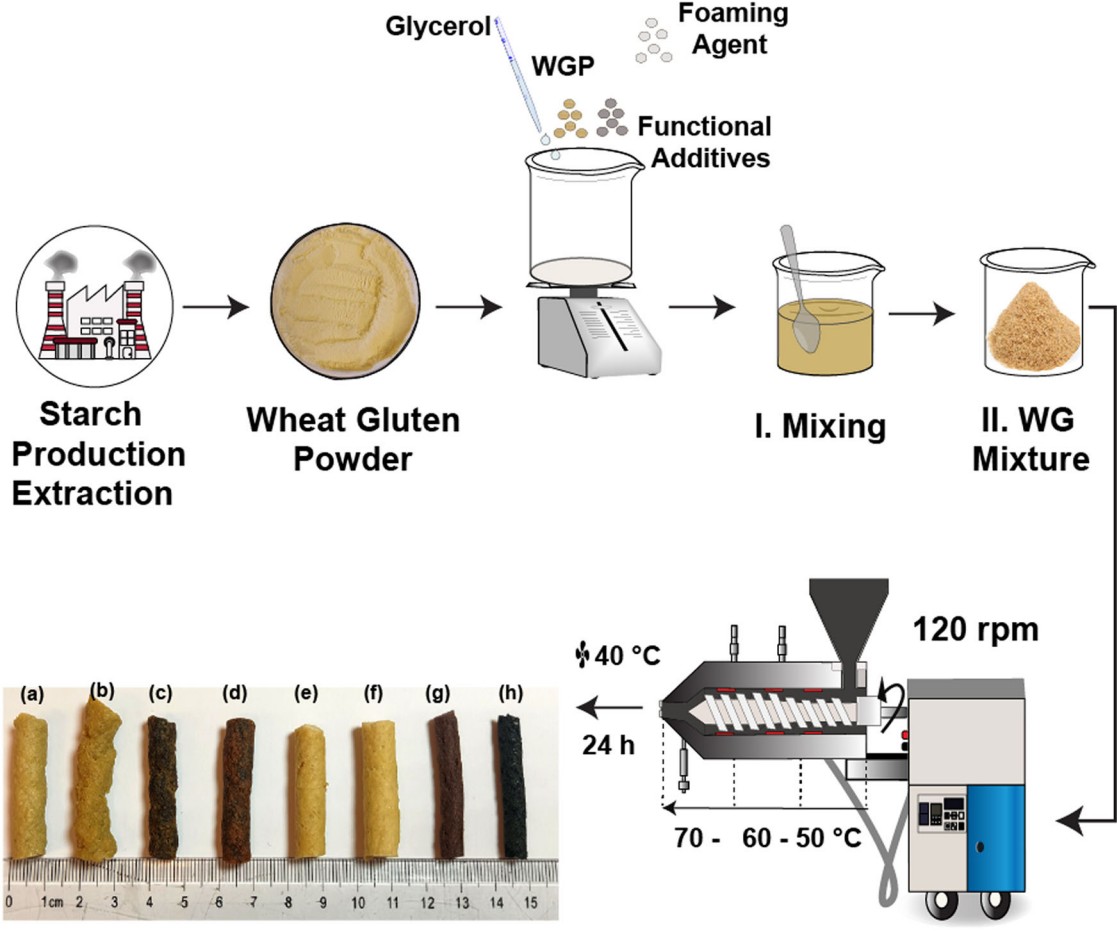

**Fig. 1 | Experimental protocol used for wheat gluten foam preparation.** Description from the extraction of the protein, mixing with the additives (I and II), extrusion of the foams (III) and appearance of the final product (IV): **a**, WG/G, **b**, WG/G/ABC, **c**, WG/G/ABC/1GA, **d**, WG/G/ABC/5GA, **e**, WG/G/ABC/1CA, **f**, WG/G/ABC/5CA, **g**, WG/G/ABC/1GNP and, **h**, WG/G/ABC/5GNP.

**Table 1 | Physical properties of the different extrudates**

| Samples | Density (kg/m³) | Expansion ratio (ER) | Open porosity (%) | Closed porosity (%) | Total porosity (%) | Pore size (µm)[1] | DC (%) |
|---|---|---|---|---|---|---|---|
| WG/G solid material | 1290 [g] | - | - | - | - | - | - |
| WG/G | 883 ± 2[e] | 1.45 ± 0.04[d] | 1.9 ± 0.1[a] | 29.7 ± 0.3[d] | 31.6 ± 0.2[b] | 65 ± 30[a] | 0[b] |
| WG/G/ABC | 720 ± 7[b] | 1.42 ± 0.09[d] | 7.7 ± 0.5[b] | 36.4 ± 0.9[e] | 44.1 ± 0.6[d] | 215±159[ab] | 13[b] |
| WG/G/ABC/1GA | 840 ± 20[d] | 1.21 ± 0.02[c] | 16.4 ± 2.6[d] | 18.6 ± 0.7[c] | 35.0 ± 1.8[c] | 145±80[ab] | −8[b] |
| WG/G/ABC/5GA | 820 ± 20[cd] | 1.24 ± 0.02[c] | 20.0 ± 2.2[d] | 16.7 ± 1.1[b] | 36.7 ± 1.8[c] | 190±100[ab] | −121[a] |
| WG/G/ABC/1CA | 641 ± 6[a] | 1.46 ± 0.07[d] | 10.9 ± 1.4[c] | 39.3 ± 1.8[f] | 50.2 ± 0.5[f] | 195±116[ab] | − 2[b] |
| WG/G/ABC/5CA | 650 ± 3[a] | 1.60 ± 0.03[e] | 7.5 ± 0.4[b] | 41.9 ± 0.4[g] | 49.4 ± 0.2[e] | 183 ± 69[b] | − 3[b] |
| WG/G/ABC/1GNP | 950 ± 40[f] | 1.00 ± 0.05[b] | 11.8 ± 3.5[cd] | 14.3 ± 1.4[b] | 26.1 ± 2.9[a] | 190±108[ab] | - |
| WG/G/ABC/5GNP | 804 ± 10[c] | 0.86 ± 0.01[a] | 26.5 ± 1.8[e] | 11.0 ± 1.3[a] | 37.5 ± 1.1[c] | 154 ± 55[b] | - |

DC: Degree of cross-linking of the different foams at 70 °C. The DC values were normalized to that of the WG/G reference. Note: Different letters (a–g) mean that the values are significantly different ($P < 0.05$) in each column. Supplementary Fig. S4 shows the distribution of the pore sizes associated with the standard deviations presented here.
[1]The pore size distribution was obtained from measurements based on SEM images; values and standard deviations (±1 standard deviation) were based on a minimum of 50 measurements on each sample.

allowing a greater chain disentanglement/molecular relaxation in the extruder. The latter may also increase pore collapse after the die. Radical scavenging can prevent or delay protein aggregation, possibly leading to a reduced or absent die-swell (less intermolecular disulphide bridges)[32]. To investigate this further, foams with a larger content of additives (10 wt%) were produced. For genipin, the ER was the same as with 5 wt%, but for gallic acid and citric acid, the ER decreased to 0.7. The foam became softer at this higher content, indicating that GA, CA and GNP are also plasticizers. However, at 1 and 5 wt.%, the content is too low to show a noticeable plasticization in the current systems.

Supplementary Table S1 shows the amount of gas generated when ammonium bicarbonate blowing agent was decomposed at 70 °C and 1 atm pressure. The theoretical volume of gas formed from the 50 g sample was 2530 mL, significantly higher than the experimentally obtained gas volume after 10 min (30 mL, Supplementary Table S1, Supplementary Video S1). Hence, the decomposition of ABC was probably not complete during the extrusion. Observe that the pressure exerted by the latex on the gas would also limit the gas volume formed, however, not likely yield the very large difference between calculated and measured gas volume. In addition, to validate the true content of gas formed in the process, pure ABC was heated separately in a sealed bag for 10 and 20 min, respectively. The results revealed that the amount of gas released from pure ABC was higher (with a gas descomposition rate, DR: 20 mL/min) compared to WG/G/ABC (DR = 4 mL/min calculated during the first 20 min), but still lower than the theoretical value. In order to increase the extent of ABC decomposition, a temperature increase is needed. If the temperature increases from 70 to 90 °C, the decomposition rate increases substantially; nevertheless, a 30 min period was needed to obtain a 90% decomposition (Supplementary Fig. S2). The higher efficiency of the foaming process at elevated temperatures must be weighed against the increasing aggregation, and disulphide intermolecular crosslinking, of the protein at higher temperatures. This was a contributing reason to the poor foaming when SBC was used[18].

The reference WG/G foam structure was similar to that presented previously[18], consisting mainly of small ~65 µm closed cells (Fig. 2a and Supplementary Fig. S3a, Table 1), originating from the porosity-generating expansion of the moisture-generated steam. It also had the narrowest pore-size distribution (Supplementary Fig. S4a). Although the materials were dried before extrusion, the small WG powder particles quickly absorbed moisture in the hopper. The density of this foam was 883 kg/m³, corresponding to a porosity of 32% (2% open and 30% closed porosity). With the use of ammonium bicarbonate (WG/G/ABC), the density decreased to 720 kg/m³, and the open and closed cell porosity increased to 8 and 37%, respectively (Fig. 2b, Supplementary Fig. S3b and Table 1). The average cell size was also considerably larger, with an average pore size of *ca.* 215 µm, compared to *ca.* 65 µm for the reference material. The surface of the WG/G/

ABC extrudate was rougher compared to the other foams (Fig. 1), which was most likely due to the partial collapse of the cells when exiting the die[18].

The use of gallic acid and genipin resulted in higher densities/lower porosity (Fig. 2c, d, g, h and Supplementary Figs. S3c, d, g, h, Table 1), which was in marked contrast to when the citric acid was used (Fig. 2e, f). Interestingly, the use of citric acid, in combination with ABC, yielded the lowest density (641–650 kg/m³) and consequently the highest porosity (50-49%) with the prevalence of closed cells. The crosslinking with citric acid is suggested to occur through citric anhydride that reacts with amines in the protein (Supplementary Fig. S5a)[33]. Citric acid can also react with ammonium bicarbonate at ambient conditions to yield ammonium citrate, releasing carbon dioxide and water, and contributing to pore formation[44,45]. The density was not significantly different at 1 and 5 wt% CA. This, together with SE-HPLC data and degree of crosslinking presented below, indicated a similar extent of aggregation and crosslinking in the two foams.

The surface regions were often denser than the interior due to the pressure exerted by the barrel wall on the material, which is likely the reason for the radial gradient in colour observed in Supplementary Fig. S1. The variation in foam density was most easily observed with SEM in the foams with the lowest density (citric acid). As revealed from the cross-sectional view (Supplementary Fig. S6), the porosity appeared greater than indicated by the density data (Table 1). The SEM image taken along the long axis of the extrude (Supplementary Fig. S7) shows, however, a porosity that reflects better the measured ~50% porosity (Table 1).

The largest content of open cells was observed for the 5 wt% GNP system (Fig. 2h and Supplementary Fig. S4h, Table 1). A relatively high density was observed in the presence of gallic acid (Table 1). Gallic acid can crosslink through its acidic hydrogen and three hydroxyl groups in an alkaline medium, turning it into its reactive quinone form (the pH in the mixture during the processing was 7.50 and 8.50 for WG/G/ABC/1GA and WG/G/ABC/5GA, respectively)[26]. The quinone can react with amine and sulfhydryl groups in the protein[42,46], forming covalent C-N and C-S bonds, which results in crosslinked/polymerized species (Supplementary Fig. S5b)[47,48]. It has also been observed that GA, at low concentrations, decreases carbonyl formation and protects free amine. At higher concentrations, however, GA causes a high loss of sulfhydryls[26]. As with CA, the density and porosity were similar at 1 and 5 wt% GA. However, when immersing the WG/G/ABC/GA samples in water, the liquid turned brown, with a darker foam at the higher content of GA. This indicated that GA, at least at the higher concentration, acted mainly as a radical scavenger, hosting the sulphur-originating radical inside the aromatic ring and delaying the disulphide crosslinking[32], yielding more dissolvable GA-containing oligomers/peptides in the sample. These results are in line with the SE-HPLC, and degree of crosslinking, results presented below.

The 1 wt% genipin sample (WG/G/5ABC/1GNP) had a higher density (950 kg/m³) than the 5 wt% genipin sample (WG/G/ABC/5GNP, 804 kg/m³)

**Fig. 2 | Low magnification SEM micrographs of the foams. a**, WG/G, **b**, WG/G/ABC, **c**, WG/G/ABC/1GA, **d**, WG/G/ABC/5GA, **e**, WG/G/ABC/1CA, **f**, WG/G/ABC/5CA, **g**, WG/G/ABC/1GNP and, **h**, WG/G/ABC/5GNP.

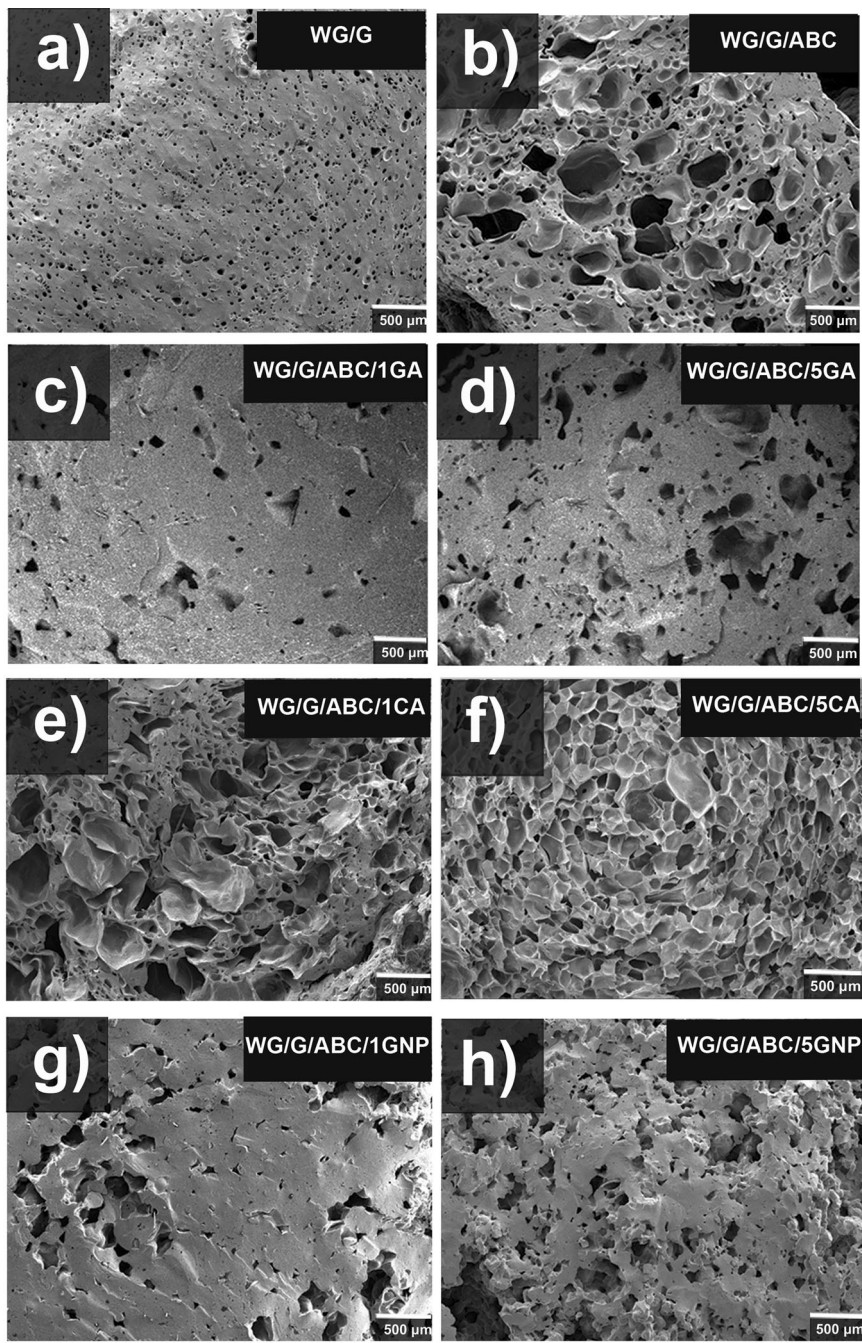

and a lower total porosity (26 versus 38%). In fact, the 1 wt% genipin foam had the highest density and lowest porosity of all samples. This can be explained by the reaction of genipin with wheat gluten (Supplementary Fig. S5c). Genipin is a bicyclic compound with complex and not fully understood reaction mechanisms[49]. It is suggested that it reacts with proteins through primary amine groups to form covalently crosslinked networks[20]. It is considered that the crosslinking process goes via two reactions on different sites on the genipin molecule. The fastest, and therefore first reaction, begins with the nucleophilic attack on the genipin C3 carbon by the primary amine group (Supplementary Fig. S5c, reaction I). This reaction leads to crosslinked networks with dimer, trimer, and tetramer bridges[49]. The slow second reaction occurs between the secondary amine in the protein with the ester group on the genipin molecule (Supplementary Fig. S5c, reaction II)[20,21,50]. The extent of these reactions depends on the amount of crosslinker added (1 wt% and 5 wt% used here), time of the reaction and the pH[49,50]. According

to Butler et al.[21], reactions I and II proceed through acid catalysis. However, according to Mi et al.[49], the ring-opening of the genipin molecule and subsequent reaction with the primary amino group occurs at basic (pH 9.0) conditions. The pH of the WG material during the foaming process was 8.8-9.0, hence it is probable that the genipin reactions occurred primarily by reaction I. At high pH, genipin is also known to self-polymerize in parallel to crosslinking with the protein, yielding large chains of crosslink bridges in the protein (Supplementary Fig. S5d)[49]. This self-polymerization my reduce the aldehyde groups in ring-opened genipin chains, resulting in a decrease in crosslinking between polymeric genipin and protein molecules, yielding possibly a larger matrix expansion (a higher porosity and lower densities)[49,51]. When immersing the GNP samples in water, a more intense blue colour of the solution was observed for the 5 wt% samples compared to the 1 wt% sample. The total solubility in the SE-HPLC measurement (discussed below) was also higher in the 5 wt% foam. Hence, a higher content of genipin-

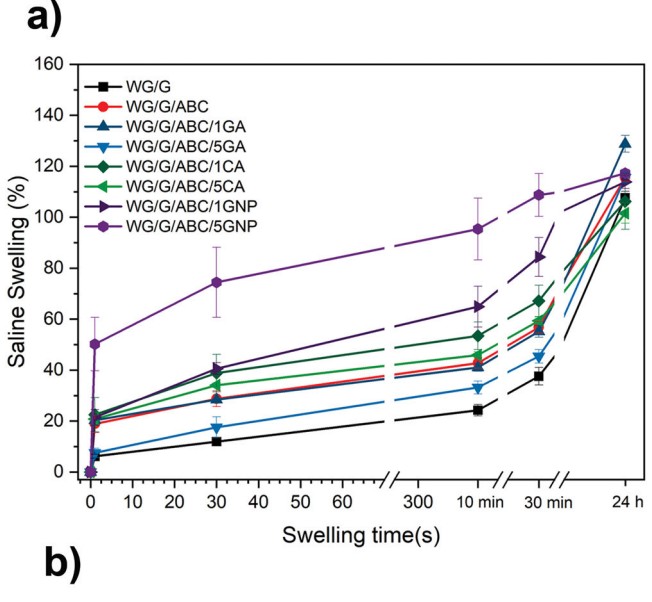

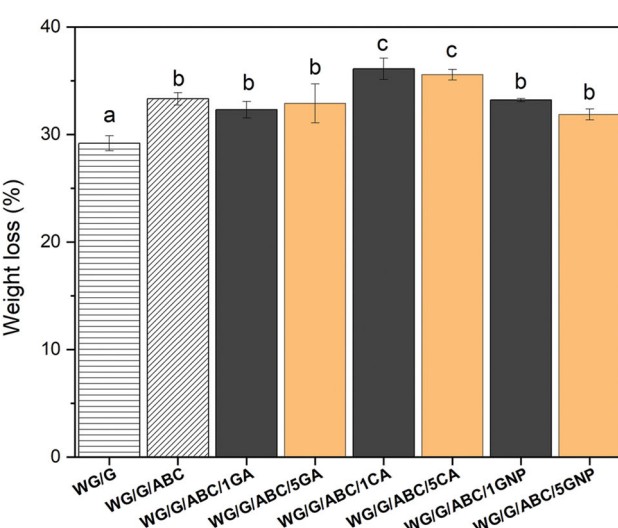

**Fig. 3 | Swelling results. a**, Uptake capacity of the WG foams in saline solution, **b**, WG foam weight loss after 24 h immersion. Error bars represent ± 1 standard deviation. Note: Different letters (a–c) means that the values are significantly different (P < 0.05).

containing oligomers, peptides and polypeptides were present at the higher genipin content. The fact that a "blue" substance came out from the foam in the solution, indicated that genipin also yielded grafted (not crosslinked) protein molecules.

## Liquid swelling capacity

The swelling/uptake capacity (SC) in saline solution is presented in Fig. 3a. The reason for using saline is that it is a body fluid model substance, and a possible application for the foams is as the absorbent core in disposable sanitary products[20]. All samples showed a rapid (capillary) uptake (within 1 s) with the highest uptake for WG/G/ABC/5GNP (SC: 50%), which was the foam with the highest content of open pores (Table 1). The lowest saline uptake from 1 s to 24 h was observed for WG/G (107%), with the lowest open pore content and the smallest pore sizes. Moreover, the highest final uptake (24 h) was observed for the WG/G/ABC/1GA foam, probably due to a combined effect of high polarity, since GA contains multiple hydroxyl groups, and the low degree of crosslinking. Finally, it can be concluded that the 24 h uptake was not significantly different between WG/G/ABC/1CA (ca. 106%) and (WG/G/ABC/5CA) (ca. 101%).

Along with the uptake of saline, there was a continuous loss of glycerol from the sample to the saline solution (Fig. 3b)[52,53]. Hence, the reported swelling values in Fig. 3a were in reality significantly larger than presented. The weight loss was in most cases higher than 30 wt%, which indicated that species related to the additives were also extracted by the saline solution (Fig. 3b)[52,53].

## Mechanical properties

The mechanical properties are reported in Table 2 and examples of compression stress-strain curves are given in Fig. 4a. The area enclosed by the hysteresis loop corresponds to the energy loss rate. At 50% strain, the WG/G sample showed the highest compression stress (1.7 MPa), followed by the gallic acid systems (WG/G/ABC/1GA: 1.3 MPa, WG/G/ABC/5GA: 1.3 MPa). After adding 5 wt% citric acid, the compression stress at 50% strain decreased considerably (down to ~0.5 MPa), due to the high porosity obtained ( ~ 50%, Tables 1 and 2). The highest stiffness (independent of the final strain (10-50%)) was observed for the 1% genipin sample, which was also the densest foam (WG/G/ABC/1GNP). The energy loss rate for all samples was comparable and their values at 50% strain ranged between 84% (5 wt% CA) and 93% (5 wt% GA) (Table 2). These high values indicate that the foams dissipate a large amount of energy, possibly acting as dampers in low-velocity impact applications[54]. The energy loss rate also increased somewhat with increasing strain and number of cycles (Fig. 4a, e.g., for WG/G, it went from 85% in the first cycle to 87% in the fifth cycle. It should be mentioned that with the 5 min relief of stress between each cycle, the strain did not go back fully, and this was more pronounced with increasing number of cycles, as well as when the strain was kept constant (Fig. 4a, image

**Table 2 | Mechanical properties at different compression strains**

| Samples | $\sigma_{y10\%}$[1] (kPa) | $\sigma_{10\%}$[2] (kPa) | $\sigma_{30\%}$ (kPa) | $\sigma_{50\%}$ (kPa) | $E_{10\%}$[3] (MPa) | $E_{30\%}$ (MPa) | $E_{50\%}$ (MPa) | $A_{f10\%}$[4] (%) | $A_{f30\%}$ (%) | $A_{f50\%}$ (%) |
|---|---|---|---|---|---|---|---|---|---|---|
| WG/G | 52±24[bc] | 259 ± 34[c] | 727 ± 44[d] | 1675 ± 64[e] | 4.1 ± 1.7[d] | 3.6 ± 0.6[d] | 0.04 ± 0.02[b] | 84.8 ± 0.2[bc] | 87.1 ± 0.7[b] | 87.1 ± 0.8[b] |
| WG/G/ABC | 18 ± 9[a] | 108 ± 8[b] | 308 ± 26[b] | 721 ± 102[b] | 0.78 ± 0.33[a] | 0.28 ± 0.01[a] | 0.02 ± 0.01[b] | 83.3 ± 1.6[abc] | 85.2 ± 0.7[a] | 87.5 ± 0.5[b] |
| WG/G/ABC/1GA | 17 ± 8[a] | 120 ± 24[b] | 355±92[bc] | 1323±709[abcd] | 0.35 ± 0.32[a] | 0.31 ± 0.29[a] | 0.04 ± 0.01[b] | 82.4 ± 0.7[a] | 85.5 ± 0.8[a] | 88.3 ± 1.5[b] |
| WG/G/ABC/5GA | 63 ± 2[c] | 221 ± 8[c] | 583 ± 50[c] | 1292 ± 152[c] | 4.1 ± 0.5[d] | 3.2 ± 0.1[d] | 0.008 ± 0.001[a] | 87.3 ± 0.1[d] | 90.2 + ± 0.1[d] | 92.5 + ± 0.4[d] |
| WG/G/ABC/1CA | 30±8[ab] | 109 ± 13[b] | 314 ± 5[b] | 723 ± 25[b] | 2.0 ± 0.5[bc] | 1.7 ± 0.3[c] | 0.02 ± 0.01[b] | 81.4 ± 0.5[a] | 84.7 ± 0.1[a] | 86.6 ± 0.1[ab] |
| WG/G/ABC/5CA | 20 ± 5[a] | 72 ± 16[a] | 216 ± 14[a] | 534 ± 82[a] | 1.4 ± 0.2[b] | 1.0 ± 0.1[b] | 0.11 ± 0.05[c] | 83.4 ± 0.7[b] | 84.7 ± 0.7[a] | 84.2 ± 1.2[a] |
| WG/G/ABC/1GNP | 60 ± 2[c] | 192 ± 38[c] | 530 ± 105[c] | 1212 ± 238[d] | 4.4 ± 0.3[d] | 6.7 ± 0.7[e] | 0.19 ± 0.10[c] | 84.2 ± 0.5[bc] | 87.3 ± 0.1[b] | 89.1 ± 1.2[bc] |
| WG/G/ABC/5GNP | 30 ± 3[b] | 96±10[ab] | 319 ± 3[b] | 902 ± 60[c] | 2.0 ± 0.1[c] | 3.4 ± 0.6[d] | 0.23 ± 0.19[c] | 85.0 ± 0.4[c] | 87.9 ± 0.1[c] | 90.4 ± 0.2[c] |

Note: Different letters (a–e) within each column mean that the values are significantly different (P < 0.05).
[1]$\sigma_{y10}$%: Yield strength at 10% of deformation.
[2]$\sigma_{10}$%: Maximum compression strength at 10% deformation.
[3]$E_{10}$%: Elastic modulus at 10% deformation.
[4]$A_{f10}$%: Hysteresis loss rate at 10% deformation.

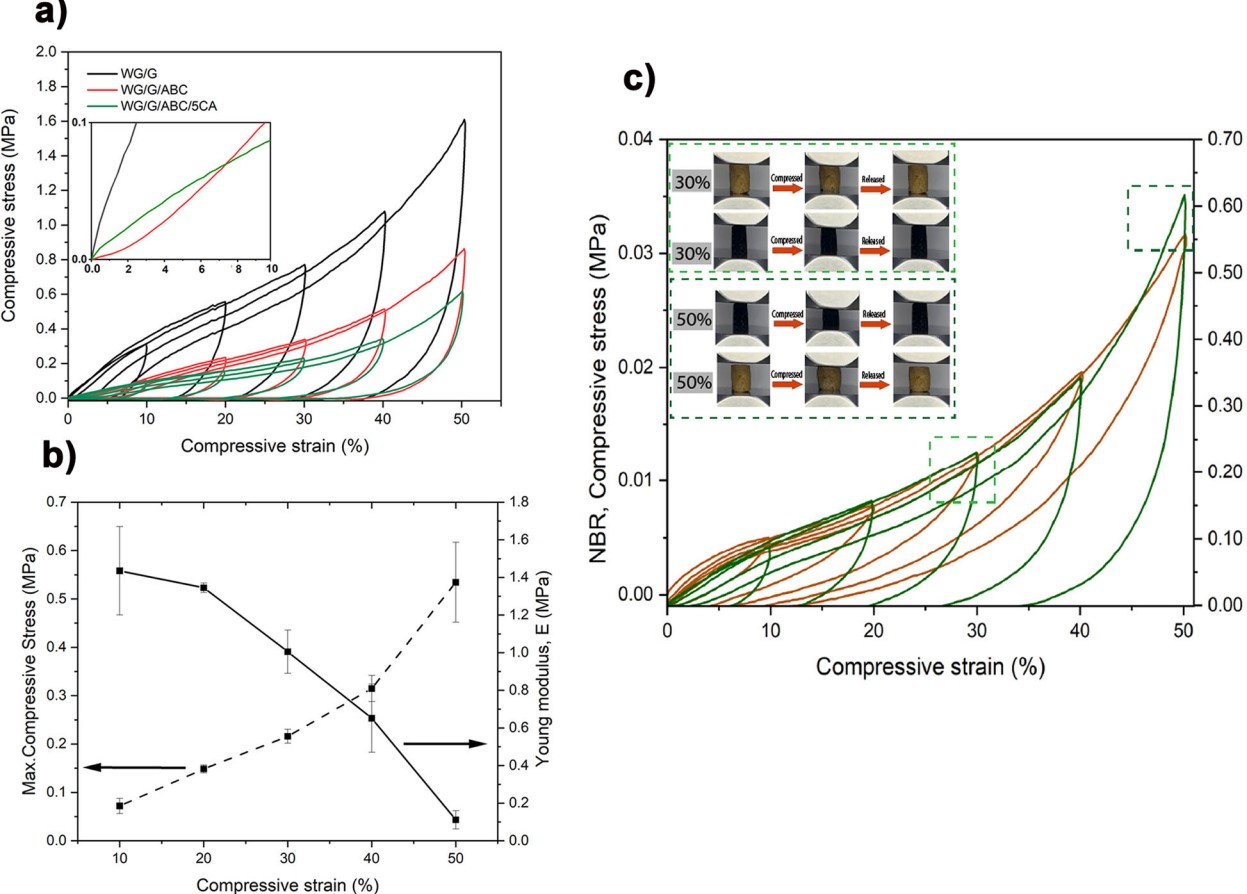

**Fig. 4 | Mechanical cycling results. a** Compression curves for the WG/G, WG/G/ABC, and WG/G/ABC/5CA foams, inset showing the 0-10% strain region. **b** The compression strength (dashed curve) and elastic modulus (solid curve) of the WG/G/ABC/5CA foam as a function of compressive strain. Error bars represent ± 1 standard deviation. **c** Compressive stress versus compressive strain of the NBR (brown curves) and WG/G/ABC/5CA (green curves) samples. Dashed squares represent different behaviors at 30 and 50% strain. Inset shows the appearence of the samples (NBR is the black sample) before and after being compressed to 30 and 50%.

inset in Fig. 4c). Because of the increasing densification with increasing strain, the compression stress increased monotonically with increasing strain (Fig. 4b, Table 2). On the other hand, the stiffness/elastic modulus decreased with increasing strain and number of cycles, indicating some non-reversible damage to the foam structure. This was probably occurring at a low extent, and was difficult to observe (Supplementary Fig. S8), which also explains the larger energy loss rate with increasing strain/number of cycles. This decrease in elastic modulus was observed for all foams.

The mechanical behavior of the 5 wt% citric acid foam during the cycling was compared to that of a commercial nitrile butadiene rubber (NBR) foam (Fig. 4c and Table 3). The reason to compare with NBR is that it is more polar/hydrophilic than most other rubbers and thus has a chemistry closer to that of WG[55,56]. It is used in, e.g., tapes, strips, sheets, seals, and heat and sound insulations[57,58]. The increase in stress in consecutive strain cycles was similar for the two materials. However, the size of the stress and stiffness was significantly smaller for the NBR, since its density (120 kg/m³) was lower than that of the WG/G/ABC/5CA sample. The energy loss rate (refer to $A_{f50\%}$ in Table 3) was also lower for NBR. However, the elastic modulus increased with increasing strain and number of cycles, unlike in the WG/G/ABC/5CA case, indicating no significant damage to the NBR foam structure.

Repeated compression to 30% strain revealed overall essentially a full strain recovery during 5 cycles and consequently no significant change in energy dissipation for the WG foams (WG/G and WG/G/ABC/5CA, $A_f$ of ~80%) as well as for the NBR foam ($A_f$: ~55%) (Supplementary Fig. S9 and Table S2). However, a decrease in the modulus for both WG samples

indicated a low degree of softening during the 5 cycles. However, whereas the strength also decreased somewhat with increasing number cycles for the denser WG/G foam, the opposite was observed for the less dense 5 wt% CA foam. The stiffness and strength did not change during the cycling of the NBR foam.

The compression set and strain recovery are displayed in Fig. 5. The degree of recovery after the compression set at 40% strain, decreased with increasing compression time (from 1 h up to 1 week) for the reference (WG/G) (Fig. 5a) and the 5 wt% CA (Fig. 5b) samples, as well as the NBR samples (Fig. 5c). After a compression for 1 h, WG/G recovered to 96% (48 h after the compression test (Fig. 5a)) and to 99% one month after the compression test (Supplementary Table S3). The corresponding values for WG/G/ABC/5CA were 94% (Fig. 5b) and 97% (Supplementary Table S3). However, the degree of plastic deformation increased when the samples were subjected to a longer compression time (CT) (Fig. 5a–c. and Table S3). The plastic/creep deformation was slightly greater for the WG/G/ABC/5CA foam (Fig. 5a,b, and Supplementary Table S3). The reason was probably that the thinner cell walls in the latter sample showed less resistance to plastic deformation. Hence, the compression set of the reference sample WG/G was lower (31% after 1 week) than that of the foam sample with citric acid, WG/G/ABC/5CA (40% after 1 week) (Fig. 5d).

The NBR foam showed a higher compression set after longer periods than the WG foams (up to 84% vs. ~ 40%, after 48 h, Fig. 5d), indicating a lower resistance to permanent deformation under a given strain. The specific NBR foam used here is used as a material in training mats, which shows that WG-based foams have the potential to be used in similar applications.

**Table 3 | Mechanical properties of WG/G/ABC/5CA and NBR foams at 10, 30 and 50% compression strain**

| Samples | [1]$\rho$ (kg/m³) | [2]$\sigma_{y10\%}$ (kPa) | [3]$\sigma_{10\%}$ (kPa) | $\sigma_{30\%}$ (kPa) | $\sigma_{50\%}$ (kPa) | [4]$E_{10\%}$ (MPa) | $E_{30\%}$ (MPa) | $E_{50\%}$ (MPa) | [5]$A_{f10\%}$(%) | $A_{f30\%}$ (%) | $A_{f50\%}$ (%) |
|---|---|---|---|---|---|---|---|---|---|---|---|
| **NBR foam** | 120 ± 8ᵃ | 2.0 ± 0.3ᵃ | 31 ± 20ᵃ | 43 ± 30ᵃ | 105 ± 70ᵃ | 0.09 ± 0.01ᵃ | 1.3 ± 0.1ᵇ | 1.3 ± 0.3ᵇ | 67.0 ± 5.0ᵃ | 60.6 ± 6.0ᵃ | 52.0 ± 8.0ᵃ |
| **WG/G/ABC/5CA** | 650 ± 3ᵇ | 20 ± 5ᵇ | 72 ± 16ᵇ | 216 ± 14ᵇ | 534 ± 82ᵇ | 1.4 ± 0.2ᵇ | 1.0 ± 0.1ᵃ | 0.11 ± 0.05ᵃ | 83.4 ± 0.7ᵇ | 84.7 ± 0.7ᵇ | 84.2 ± 1.2ᵇ |

Note: Different letters mean the values are significantly different ($P < 0.05$) in each column.
[1]$\rho$: Density
[2]$\sigma_{y10\%}$: Yield strength at 10% deformation.
[3]$\sigma_{10\%}$: Maximum compression strength at 10% deformation.
[4]$E_{10\%}$: Elastic modulus at 10% deformation.
[5]$A_{f10\%}$: Hysteresis loss rate at 10% deformation.

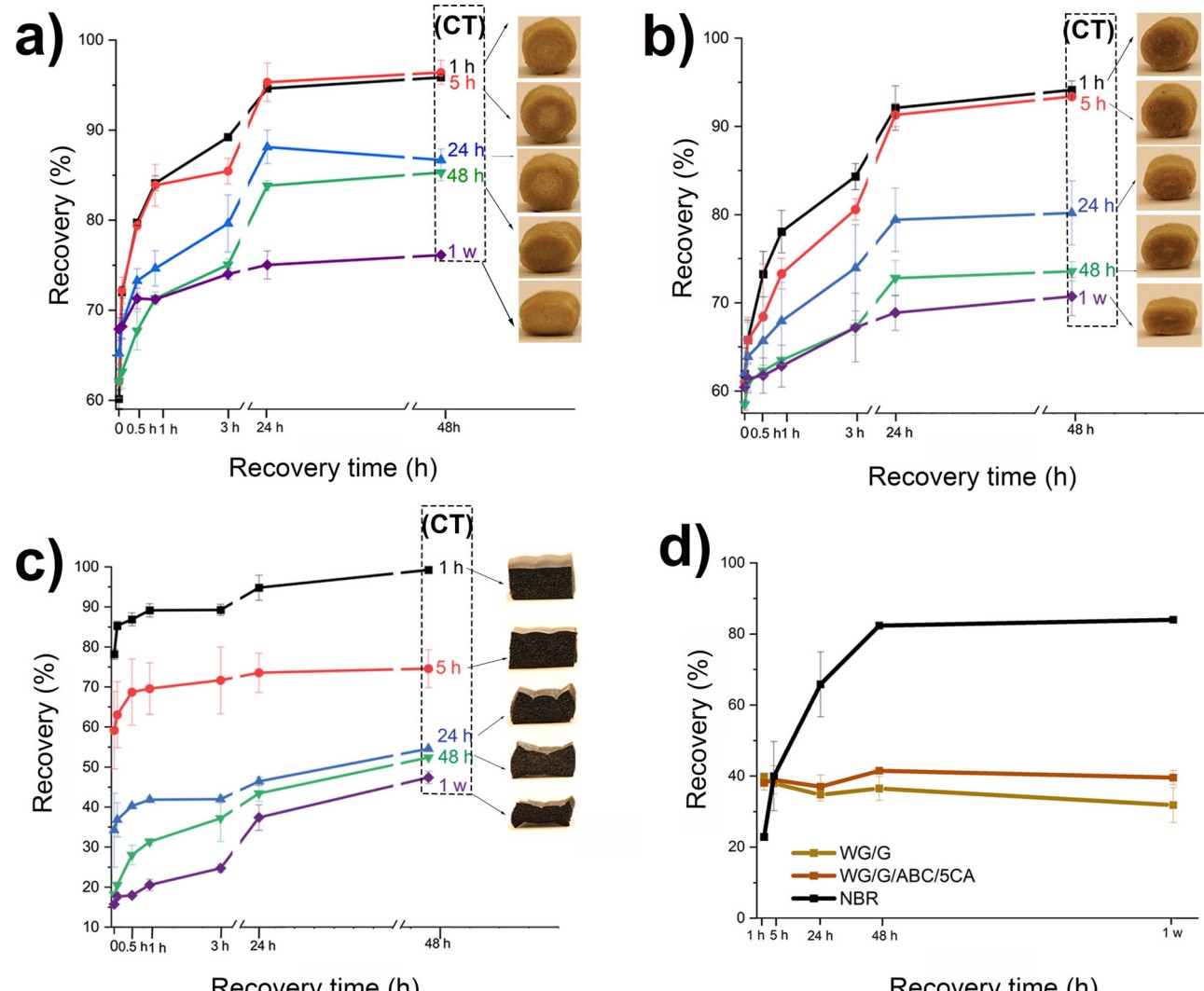

**Fig. 5 | Mechanical compression set results.** Recovery of the samples as a function of time after the release of the compressive strain, and their respective appearance after a compression time (CT) of 1 h to 1 week: **a**, WG/G, **b**, WG/G/5ABC/5CA, and, **c**, nitrile butadiene rubber. **d**, Compression set values just after the release (1 s) from the compression in the compression set testing device. Error bars represent ± 1 standard deviation and CT means compression time.

## Molecular structure and reactions, as revealed by FTIR and proton NMR

The full FTIR spectra of the samples are shown in Supplementary Fig. S10a, with a magnification of the region of special interest here. In the region 800–1150 cm⁻¹, bands originating from vibrations associated with C–C and C–O bonds/O-H deformation in glycerol were observed, with expected bands at ~ 850, 920, 995, 1030, and 1104 cm⁻¹ [59,60]. All these bands were found in all samples with glycerol. The amide I band (region centered around ~1630 cm⁻¹) represents the C = O stretch vibration coupled to N-H bending vibration [61]. The spectra of the samples in this region differed from that of the WG powder (Supplementary Fig. S10b). The fact that they peaked in the 1700-1640 cm⁻¹ region, indicated that they contained a sizeable fraction of α-helices (1660-1640 cm⁻¹) and unordered/random coil polypeptides as shown previously [62]. This, in turn, indicated a low protein aggregation (low degree of polymerization and crosslinking), which was expected due to the low extrusion temperature used (≤ 70 °C) [6,8,63,64]. Note

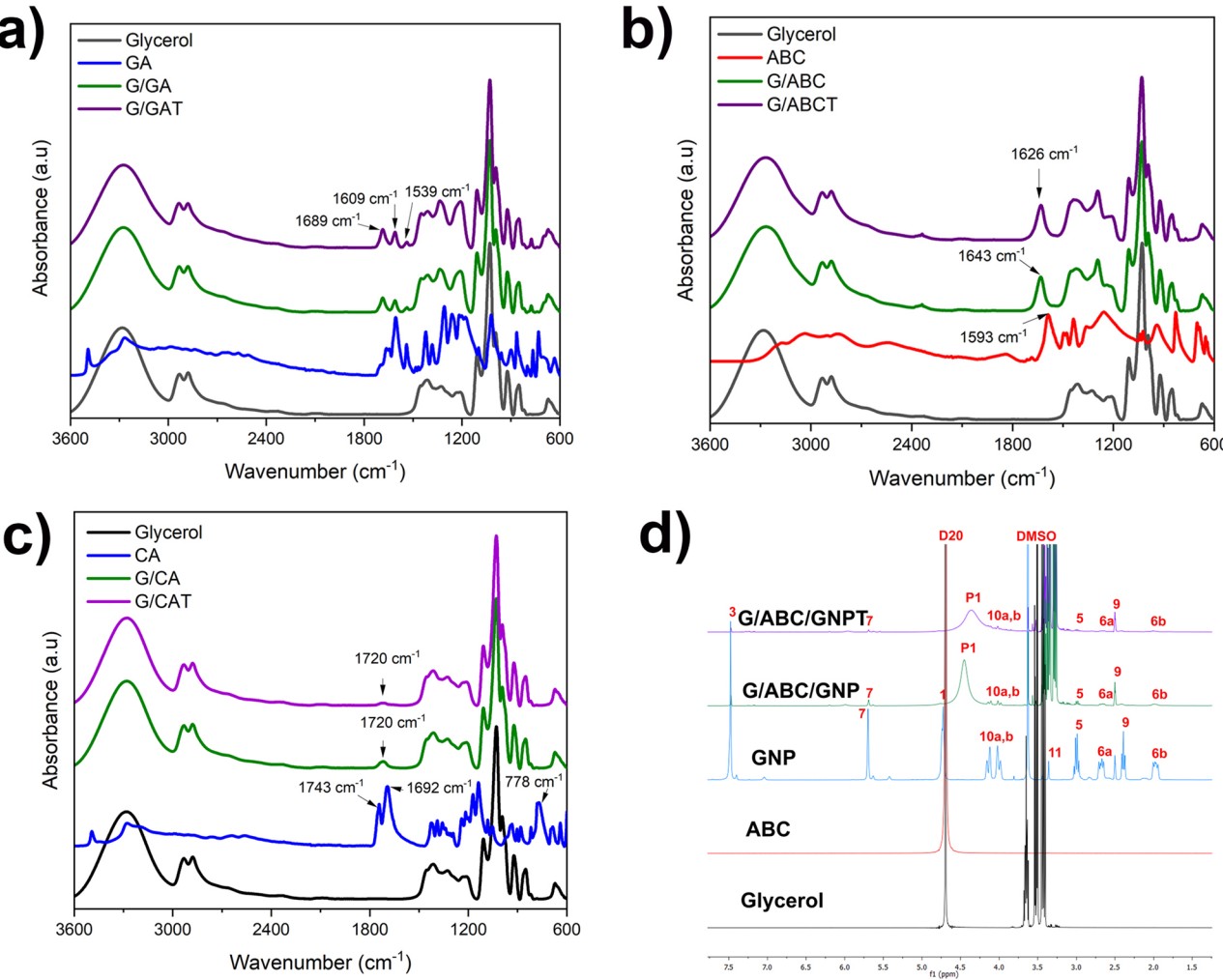

**Fig. 6 | FTIR and $^1$H NMR spectra of the pure and combined additives. a**, G/GA, **b**, G/ABC, **c**, G/CA and, **d**, $^1$H NMR spectra of the pure and combined additives in the genipin system.

here that the additives have absorbance bands in this region (Supplementary Fig. S10d), however, due to their low content they were not expected to influence significantly the general spectral shape[65].

Additionally, differences in the carboxyl band (-COOH/-COO⁻) at 1740-1720 cm$^{-1}$, amide II band (N-H bending bands coupled to C-N stretching vibrations) at ~1550 cm$^{-1}$ and the 1420-1308 cm$^{-1}$ region were observed between the samples (Supplementary Fig. S10a)[61]. In the presence of citric acid, the amide II band intensity decreased, and the absorbance increased in the 1420 cm$^{-1}$ (COO⁻) region (Supplementary Fig. S10a). This indicated a reaction between citric acid and WG[33,66,67]. The ~1745 cm$^{-1}$ band, originating from C = O stretching in ester groups, overlapped with neighboring bands, which made it difficult to assess whether esters were formed in any of the samples (Supplementary Figs. S10c,d). The band was, however, present in all systems[60]. The low FTIR absorption indicated that at the higher concentration (5 wt%), the esterification reaction between CA and WG, and the ring-opening reaction/formation of aldehydes in GNP/ WG were not complete, limiting the degree of crosslinking.

Finally, absorption in the 1390-1330 cm$^{-1}$ region, due to O-H deformation, was observed for WG/G/ABC/5GA, a consequence of the high concentration of OH in gallic acid (Supplementary Fig. S10a)[68].

To understand further the contribution of reactions/interactions between the specific additives to the overall gluten foam properties, i.e., interactions not involving the WG matrix, FTIR and NMR were used to capture any chemical changes before and after heat treatment (70 °C for 5 min) in an oven (simulating extrusion conditions) (Fig. 6, Supplementary

Figs. S11 and S12). GA, CA, and GNP were mixed with ABC and glycerol in the same ratios as in the presence of WG (Table 1 (30:5, glycerol:functional additive)).

When GA was mixed with glycerol, the "composite" FTIR spectrum was, in general, simply a combination of the two individual constituents (Fig. 6a and Supplementary Fig. S11a.1). However, no changes in the FTIR spectrum were observed after the heat treatment, despite the colour change of the sample (Supplementary Fig. S11a.1). The absence of a change in the FTIR spectrum could be due that the extent of the reaction was low enough to be below the resolution limit of the FTIR equipment and/or that the colour change did not involve the IR spectral range[69,70]. Besides the visual colour change, the decrease/disappearance of the GA hydroxyl and carboxyl $^1$H NMR signals in the presence of glycerol also indicated interactions between GA and glycerol (Supplementary Fig. S12a). Figure 6b and Supplementary Fig. S11a.2 show that the composite FTIR spectrum was not a simple combination of glycerol and ABC. The development of the peaks at 1626 cm$^{-1}$ and 1643 cm$^{-1}$ are associated with reactions involving the ammonium ion in ABC, which has characteristic bands at 1450 − 1106 cm$^{-1}$ [71]. Hence, the presence of glycerol affected ABC, which was also indicated by the colour change after the heat treatment (Supplementary Fig. S11a.2). The $^1$H NMR spectra of the glycerol/ABC system did not reveal any additional information (Supplementary Fig. S12a.2).

Interestingly, when all three components were mixed (GA, glycerol, and ABC), a different heating-induced colour change occurred (more whitish) than in the above-described systems (Supplementary Figs. S11a.1,

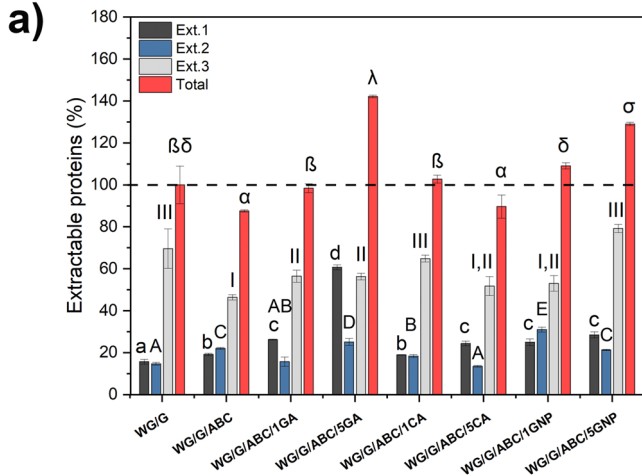

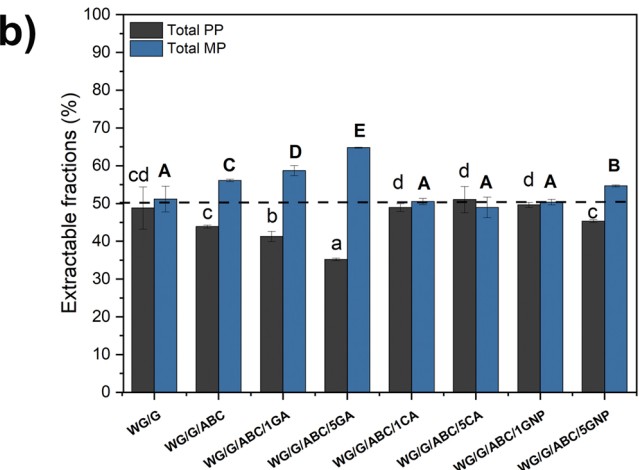

**Fig. 7 | SE-HPLC results. a** The total relative amount of extractable proteins in the three extraction steps, and, **b** relative amount of the total polymeric (PP) and monomeric (MP) proteins extracted. The total extractable protein content was normalized with respect to that of the WG/G foam processed at 70 °C. Error bars represent ± 1 standard deviation. Note: Different letters mean the values are significantly different ($P < 0.05$).

a.2 and a.3). The FTIR spectra with all three components present were too complex to be able to determine if new peaks occurred. However, the spectra were the same before and after the heat treatment.

When mixing citric acid and glycerol, the two CA peaks at 1692 cm$^{-1}$ and 1743 cm$^{-1}$ [72], ascribed to the C=O stretch in the carboxylic group merged into a single peak at 1720 cm$^{-1}$, as observed when ABC was also present (Fig. 6c and Supplementary Figs. S11b.1 and b.3)[60]. This has been ascribed to a combination of ester bond formation (here between CA and glycerol) and the presence of remaining unreacted CA carboxylic acid[71,73]. Notice also the disappearance of the strong CA 778 cm$^{-1}$ peak in the presence of glycerol, associated with CH$_2$ rocking vibrations, which indicate interactions between glycerol and CA[74]. As in the GA system, the FTIR spectra were the same before and after the heat treatment, although the mixture became whiter after heat treatment in the three-component system (Supplementary Fig. S11b.3).

In the genipin/glycerol and genipin/glycerol/ABC systems, no evident new FTIR peaks occurred after mixing and after heat treatment (Supplementary Figs. S11c.1 and c.3). The clear genipin peaks at 1676 and 1617 cm$^{-1}$ became less intense in the mixture due to genipin being diluted with the other components. This result agrees with the $^{1}$H NMR data for this mixture (Supplementary Figs S12c.1 and c.2). However, a brown colour was observed when mixing genipin with glycerol (with and without ABC)

(Supplementary Fig. S11c.1), which is ascribed to typical oxidation reactions of the genipin molecule[43]. The intensity of the brown colour increased after heat treatment, indicating further reaction. Again, these reactions must have occurred on a small scale since they were not observed with FTIR (below the detection limit). The NMR spectrum for G/ABC/GNP (Fig. 6d and Supplementary Fig. S12c.2) contained a broad signal (P1), which could be associated with the condensation reaction between a carboxylic and alcohol group, resulting in an ester and water[71]. This broad peak was also observed in previous works where genipin was used to crosslink wheat gluten[75,76].

The SE-HPLC data showed that WG/G had the lowest protein solubility after the cleavage of only secondary bonds (Ext.1), indicating a high degree of aggregation/disulphide crosslinks in this sample (Fig. 7a). In contrast, the sample with 5 wt% gallic acid (WG/G/ABC/5GA) showed the highest solubility in Ext.1, and high total solubility/extractability, hence being the least aggregated/crosslinked among the different formulations. The combination of SDS and sonication, the latter breaking disulphide bonds (Ext.2 and Ext.3) increased the solubility of all samples, and the effect was strongest for the 5 wt% genipin foam. Consequently, the total solubility of this sample was high, only surpassed by the 5 wt% GA foam. The lowest total protein solubility was observed for WG/G/ABC and WG/G/ABC/5CA samples, revealing a high degree of aggregation/crosslinking in these samples (Fig. 7a). It is worth mentioning that the crosslinking effect could also be underestimated to some extent here, as previous work has shown that the maximum UV absorbance in gluten-based systems can sometimes be shifted from the normally used absorbance signal at 210 nm to lower wavelengths[77,78].

The amount of monomeric protein was higher than polymeric protein (Fig. 7b), except for the WG/G/ABC/5CA sample. The largest difference in soluble monomers and polymers was observed for the 5 wt% gallic acid sample (65% to 35%), showing that the high protein solubility of WG/G/ABC/5GA was due to the release of shorter protein chains without significant intermolecular disulphide crosslinking.

The degree of crosslinking using Lowry's method is presented in Table 1. The lowest crosslinking density was observed for the 5 wt% gallic acid sample (−121%, relative to the reference WG/G material) in line with the HPLC results. As pointed out earlier, gallic acid can act as a crosslinker[19,26,40], but also act as a radical scavenger and reduce the degree of protein aggregation/crosslinking, delaying the disulphide crosslinking[26]. The latter function aligns with the SE-HPLC, demonstrating that the radical scavenging effect was stronger here than the crosslinking. Apart from the 5 wt% gallic acid sample, the crosslinking values from this method were insignificantly different from the reference sample. Values for the genipin-containing samples are not presented, since the bluish colour associated with genipin reactions interfere with the detection colour of the method.

## Conclusions

It was shown here that the structure and properties of foam-extruded wheat gluten could be significantly varied by using naturally occurring citric acid, gallic acid and genipin, combined with glycerol and ammonium bicarbonate. The densities obtained ranged between 640 and 950 kg/m$^3$, representing 26–52% porosity, categorizing them as high-density foams. The open and closed foam porosity ranged between 2 and 27 and 11 and 42%, respectively, and the mean pore size varied between 65 and 215 μm. The foams also exhibited a rapid saline uptake (between 10 and 50 wt.%), and the final 24 h uptake ranged between ~100 and 130%. The mechanical properties varied between the samples; the yield strength and initial modulus ranged between 17 and 63 kPa and 0.35 and 4.4 MPa, respectively. The foams could be repeatedly compressed from 10 to 50% strain, with an energy-loss rate between 84 and 92%, indicating that the foams can dissipate a sizeable amount of energy in low-velocity impact applications. Additionally, the compression set and strain recovery, evaluated on the reference foam (WG/G) and that with ABC and citric acid, depended on the time period under compression and the recovery time.

Different interactions between the protein and the additives were observed. Citric acid and genipin acted primarily as crosslinking/grafting

**Table 4 | Sample nomenclature**

| Sample Name | WG[1] | G[1] | ABC[2] | GA[2] | CA[2] | GNP[2] |
|---|---|---|---|---|---|---|
| WG/G | 70 | 30 | | | | |
| WG/G/ABC | 70 | 30 | 5 | | | |
| WG/G/ABC/1GA | 70 | 30 | 5 | 1 | | |
| WG/G/ABC/5GA | 70 | 30 | 5 | 5 | | |
| WG/G/ABC/1CA | 70 | 30 | 5 | | 1 | |
| WG/G/ABC/5CA | 70 | 30 | 5 | | 5 | |
| WG/G/ABC/1GNP | 70 | 30 | 5 | | | 1 |
| WG/G/ABC/5GNP | 70 | 30 | 5 | | | 5 |

[1]wt% in the WG/G sample.
[2]wt%/100 g WG/G.

agents, whereas the behavior of gallic acid indicated that it acted as a radical scavenging agent. Besides reactions/interactions with the gluten protein, colour changes indicated that these additives interacted/reacted also with glycerol and ammonium bicarbonate.

To conclude, the biofoams developed here have the potential to be used in disposable sanitary products, as revealed by the size of the uptake of the body-fluid model (saline). They also have the potential to be used in cushioning and sealing applications, as indicated by the energy-dissipation and compression set values obtained. This represents a step forward in replacing fossil-based polymer foams with alternatives based on biopolymers and, thus, towards a more sustainable consumption pattern.

## Methods

### Materials
Wheat gluten (WG) powder was supplied by Lantmännen Reppe AB. The powder consisted of 82 wt% wheat gluten protein (N x 6.25), 5.8 wt% wheat starch, 1.2 wt% lipids, 0.9 wt% ash and 6.9 wt% water. Glycerol (ACS reagent ≥99.5%), ammonium bicarbonate (ABC, $NH_4HCO_3$, ACS reagent≥98%), and gallic acid (ACS reagent ≥ 97.5 (titration)), were provided by Sigma-Aldrich, Sweden. Citric acid (ACS reagent ≥99.5%) and genipin (ACS reagent ≥ 98%, HPLC grade) were purchased from Merck Life Science, and Zhinxin Biotechnology, respectively. A nitrile butadiene rubber training mat foam (NBR, density: 120 kg/m³), was obtained from a hardware store (Jula AB, Sweden). This foam was used to benchmark the mechanical properties of the produced biofoams.

### Foam preparation
A schematic of the foam preparation is shown in Fig. 1. The received WG powder (WGP) was poured into a beaker containing glycerol and manually mixed for 5 min until a homogeneous WG-glycerol mixture was obtained, with a mass ratio of 7/3 (WG/G). 5 wt% of ammonium bicarbonate (ABC), with respect to the total mass of gluten and glycerol was added as a blowing agent to all mixtures (not the pure WG-glycerol reference). 1 wt% and 5 wt% (wt%/100 g WG-glycerol) gallic acid, citric acid, or genipin were then also added to the mixture before the extrusion. The ABC and CA were grounded using a Retsch Ultra Centrifugal Mill ZM 200 with a ring sieve labeled 0.25 at 6000 rpm before the mixing. The mixture was extruded in a single screw extruder (Brabender Do-Corder C3), with an L/D ratio of 20 and a screw compression ratio of 2.5. The heating zones were set to 50-60-70 °C from the hopper to the die to build up the ABC reaction gradually toward the die region[18]. The screw speed was 120 rpm and a circular die with a diameter of 6.5 mm was used (Video S2). The extrudates were dried overnight at 40 °C in a ventilated oven and were then stored in a desiccator containing silica gel for at least 1 week before any test (relative humidity (RH) ≤ 10%). The full description of the samples is given in Table 4. The reference samples prepared with glycerol and ammonium bicarbonate were named WG/G and WG/G/ABC, respectively, whereas the samples with the GA, CA, and GNP

additives were named e.g., WG/G/ABC/1GA, where the number refers to the GA, CA or GNP concentration.

To understand the contribution of ammonium bicarbonate in the foaming process, the pure ammonium bicarbonate powder, WG/G and WG/G/ABC mixtures were first placed in a rubber latex-sealed bag and then placed into a measuring cylinder with silicon oil. The equipment was placed in a heat-transfer oil (Mobil Therm 605). The temperature of this oil was set at 70 °C. The amount of gas released was recorded from the beginning and up to 20 min and calculated according to ref. 79 where the decomposition rate and total volume of gas generated by ammonium bicarbonate were determined by Eqs. (1) and (2).

$$\text{Decomposition rate} = \frac{\text{Volume of oil displacement}}{\text{time}} \quad (1)$$

$$\text{Total volume of gas} = \frac{\text{Volume of oil displacement}}{\text{Mass of ammonium bicarbonate}} \quad (2)$$

The theoretical volume of released gas by heating the foaming agent during the foaming was also calculated according to ref. 80 using the gas states equation:

$$p.V = n.R.T = \frac{m}{M}.R.T \quad (3)$$

$p$ is the normal pressure (101325 Pa), $V$ is the gas volume [m³], $n$ is the amount of gas [mol], $R$ is the universal gas constant (8.134 J/(K mol)), $T$ is the temperature [K], $m$ is the mass of the foaming agent (2.5 g/50 g batch (5wt.%)) and $M$ is the molecular weight of ABC, $M = 79.06$ g/mol.

### Thermogravimetric Analysis
To determine the decomposition temperature of ammonium bicarbonate as a foaming agent, the thermal properties of the extrudate samples was determined by using a Mettler- Toledo TGA/SDTA851 instrument (Leicester, England). 12.0 ± 2 mg of the compound was placed in 70 μL alumina pans. The mass loss was determined at 70, 80 and 90 °C during 30 minutes in nitrogen (flow rate of 50 mL/min).

### Density
The sample density and porosity were calculated based on a modified Archimedes principle using a three-component model. The volume of the solid matrix in the foam sample ($V_m$) was calculated from the sample weight in the air ($m_a$) (using a Mettler Toledo AL104 balance) divided by the solid gluten/glycerol density $\rho_{WG/G} = 1290$ kg/m³. The latter was obtained from:

$$\rho_{WG/G} = 1/\left(\frac{w_G}{\rho_G} + \frac{1 - w_G}{\rho_{WG}}\right) \quad (4)$$

where $w_G$ represents the weight fraction of glycerol (0.3) in the gluten/glycerol mixture. The density of solid WG ($\rho_{WG} = 1300$ kg/m³) and glycerol ($\rho_G = 1260$ kg/m³) were determined in a previous work[16]. The sample weight ($m_L$) was measured in the nonpolar solvent limonene (density $\rho_l = 842$ kg/m³), or $n$-heptane (density $\rho_h = 684$ kg/m³) for the lowest density samples, using an Archimedes unit attached to the balance. The volume of open and closed pores was calculated with Eqs. (5) and (6) by first determining the weight of the sample in air, the weight of the sample in the liquid (immersed sample, $m_i$), and directly after being removed from the liquid (wet sample, $m_w$). It was measured after only 1 s immersion in the liquid to obtain the capillary uptake. This enabled the calculation of the volume of open and closed pores:

$$V_{\text{open pores}} = \frac{m_w - m_a}{\rho_{liq}} \quad (5)$$

$$V_{\text{closed pores}} = \frac{m_a - m_i}{\rho_{liq}} - V_m \qquad (6)$$

The open and closed porosity was then obtained as the volume of open and closed pores divided by the total volume: $V_{\text{total}} = V_{\text{open pores}} + V_{\text{closed pores}} + V_m$. The density was finally calculated as: $m_a/V_{\text{total}}$.

## Scanning electron microscopy (SEM)

The morphology of the samples was analysed using a FE-SEM Hitachi S-4800 at a voltage of 3 kV and a current of 10 μA. The foamed samples were frozen by immersing them in liquid nitrogen for 1 minute and then breaking them into pieces. The cryo-fractured pieces were fixed onto aluminum specimen holders using conductive carbon tape. The samples were coated with palladium/platinum, using an Agar High-Resolution Sputter Coater (model 208RH) for 30 s. The samples' morphology of the foams compressed at the same repetitive compression strain was observed using a Hitachi Tabletop SEM (TM-1000, Japan) at a 10 kV voltage. The average cell pore size was obtained from a minimum of 50 measurements in SEM micrographs using the image analysis software Image J.

## Swelling capacity (SC)

The swelling capacity (SC) of the WG samples were measured by placing the sample of ca. 200 mg ($W_d$) into an empty teabag (to ensure no sample loss during the testing). It was then hooked to a glass rod and immersed in a beaker containing saline solution (0.9 wt% NaCl in Milli-Q water) from 1 s up to 24 h. The bags with the material were hung for 10 s and then touched gently against tissue paper for 10 s to remove excess liquid before measuring the weight. The wet sample was removed intermittently from the tea bag and weighed ($W_i$). The SC was calculated according to Eq. (7). The empty bag and extrudate were kept in a desiccator with silica gel for a minimum of 48 hours before the SC test.

$$SC\,(\%) = \frac{W_i - W_d}{W_d} \cdot 100 \qquad (7)$$

## Fourier-transform infrared spectroscopy (FTIR)

FTIR spectra were obtained with a PerkinElmer Spectrum 100, equipped with a triglycine sulfate (TGS) detector and a Golden Gate unit (Single-reflection ATR, Graseby Specac). The scanning step was $1.0\,\text{cm}^{-1}$ with a resolution of $4.0\,\text{cm}^{-1}$. 16 consecutive scans were recorded (4000 to $600\,\text{cm}^{-1}$) for each sample. The spectra were normalized to the -$CH_2$ band at $2926\,\text{cm}^{-1}$ [75]. The samples were kept in a desiccator with silica gel for at least one week before the FTIR measurements.

## $^1$H Nuclear magnetic resonance spectroscopy (NMR)

30 mg of sample was mixed with 600 μl $D_2O$ except for the GA and GNP systems where DMSO-$d_6$ was used. These solutions were placed into an NMR quartz tube and the NMR spectra were recorded at room temperature with a Bruker Avance III HD 400 MHz NMR equipment and the data was processed with MestreNova software.

## Size-exclusion high-performance liquid chromatography (SE-HPLC)

The protein solubility and the relative molecular size were evaluated by size-exclusion high-performance liquid chromatography (SE-HPLC). A three-step extraction protocol was implemented, as described in Gällstedt et al.[78] wherein 0.5 wt% sodium dodecyl sulfate (SDS) in 0.05 M $NaH_2PO_4$ was used as solvent/surfactant in combination with multiple ultrasonication steps. In Extraction 1 (Ext.1), 14 mg of the foam was suspended in 1.4 mL SDS phosphate buffer solution (pH 6) and shaken for 5 min before using a centrifugation at 2000 rpm to obtain a supernatant. In the second extraction

(Ext. 2) the residual foam pellet in Ext.1 was resuspended in 1.4 mL of SDS phosphate solution followed by 30 s ultrasonication and then centrifugation. In the third extraction (Ext. 3), the residual foam pellet from Ext. 2 was resuspended in a new SDS buffer and sonicated for a longer time (30 + 60 + 60 s). Three replicates per formulation were used and the solubility/extraction was normalized to that of the total protein extractability after the three extractions of the pure WG powder. The chromatograms were obtained by UV-detection using a Waters 996 Photodiode Array Detector at a wavelength of 210 nm[77,78]. Subsequently, the chromatographic data were divided into a polymeric fraction (elution time from 1 to 15 min) and a monomeric fraction (elution time from 15 to 26 min).

## Crosslinking degree

The degree of crosslinking was determined with the same protocol as described in ref. [81]. Briefly, a piece of the sample (2–6 mg) was immersed in a denaturing solution for 2 h (0.086 mol/L Tris base, 0.045 mmol/L glycine, 2 mmol/L ethylenediaminetetraacetic acid (EDTA) and 10 g/L SDS phosphate solution pH 6) to denature the uncrosslinked protein. Subsequently, the extractability was determined by centrifuging the solution at 10,000 g for 10 min to separate the denatured protein solution. The amount of soluble protein was determined by Lowry's method[82], using a Genesys-20 spectrophotometer (Thermo Scientific). The crosslinking density of the different foams was normalized to that of the reference WG/G sample (considered to be 0% cross-linked, hence neglecting the disulphide bridges).

## Mechanical cycle test

The hysteresis cycle test was performed on cylindrical samples placed between two compression plates (T1223-1021) with a diameter of 50 mm, using an Instron 5944 universal testing machine. The test was performed with a 500 N load cell according to the ISO 844:2007 standard. All samples were conditioned at $23 \pm 1\,°C$ and $50 \pm 2\%$ relative humidity (RH) for at least 120 hours prior to the measurement and then cut into 5 mm long cylinders. The foams were compressed consecutively to 10, 20, 30, 40, and 50% of their length. Between the compression steps the compression plates were returned fully and kept there for 5 min to allow for the foam to recover. A low-velocity deformation rate of 10 mm/min (200%/min) was used, in agreement with previous works[83–86]. The dimensions of each specimen were measured with a digital caliper prior to the compression test. The hysteresis loss rate ($A_f$) was calculated, representing the energy difference between the loading and unloading cycles, expressed as a percentage of the loading energy. The elastic modulus ($E$) was obtained from the slope below 5% strain, according to the ASTM D1621-16 standard. The compression strength ($\sigma_s$) was obtained as the stress yield when the yield occurred below 10% specimen deformation. If no deflection below 10% of compression was observed (indicating densification of the porous foam), the load was reported as stress at 10%.

## Compression set

Compression set measurements were performed according to the ASTM D395–18 standard on samples conditioned at $23 \pm 1\,°C$ and $50 \pm 2\%$ RH for at least 72 h. The compression set testing twing consisted of two plastic plates; one upper with rectangular shape $5 \times 30 \times 20\,\text{mm}^3$ and a lower circular plate of diameter 25 mm and 5 mm thickness. The foams were cut into 10 mm long specimens and placed horizontally between the two plastic plates. The samples were then compressed to 40% strain for 1, 5, 24, 48 h and one week compression time (CT). The NBR foam was cut in a rectangular shape of $15 \times 15 \times 10\,\text{mm}^3$ and compressed to the maximum extent possible in the twings used since the foam density was significantly lower than that of the WG foams. This represents a deformation to ca. 76% strain. Two compression set devices were used for each time point; a total of 10 compression set devices were prepared per formulation. After each compression time (CT), the samples were removed from the device and allowed to rest for 1 s up to 1 month at $23 \pm 1\,°C$ and $50 \pm 2\%$ RH °C to obtain the foam

recovery as a function of time. The compression set (*CS*) was calculated according to Eq. 8:

$$CS = \frac{(t_o - t_i)}{t_o} \times 100 \qquad (8)$$

where *CS* is expressed in %, and $t_o$ and $t_i$ are the initial and final sample thicknesses, respectively.

## Statistical analysis

Statistical analyses were performed with a student's t-test, evaluating the variance of the measurements ($p < 0.05$, confidence level). The test was analysed by using the software Statgraphics 18. The results were presented as mean values with standard deviations, showing the significant difference from different superscript letters ($P \pm SD^x$). A minimum of at least three replicates were used for each measurement to obtain reliable deviations.

## Data availability

The Supplementary Information includes Colour gradient of the cross-section of wheat gluten foams (Supplementary Fig. S1); Theoretical calculation of the foaming power of ammonium bicarbonate (Supplementary Table S1); Determination of gas expansion in ammonium bicarbonate as a foaming agent (Supplementary Video S1); Ammonium bicarbonate at different decomposition temperatures (Supplementary Fig. S2); High-magnification FE-SEM micrographs of the foams (Supplementary Fig. S3); Pores size distribution of the foams (Supplementary Fig. S4); Presumed schematic illustration of the mechanisms of crosslinking (Supplementary Fig. S5); Cross-section of the WG/G/ABC/5CA foam extrudate (Supplementary Fig. S6); Transverse-section (cut along the extrusion direction) of the WG/G/ABC/5CA foam extrudate (Supplementary Fig. S7); SEM images of WG/G/ABC/5CA and NBR (b, b.1) foams at 10% (a,b) and 50% (a.1,b.1) compressive strain (Supplementary Fig. S8); Representative compression curves at the same repetitive compression strain interval (0-30%) (Supplementary Fig. S9); Mechanical properties of the sample at 30% strain (Supplementary Table S2); Recovery of the samples after one month from the compression test (Supplementary Table S3); FTIR spectra the WG foams (Supplementary Fig. S10); Molecular structure, images illustrating the different reactions in the mixtures and full FTIR spectra (Supplementary Fig. S11); $^1$H NMR spectra of the different mixtures and components (Supplementary Fig. S12), and Manufacturing of Wheat gluten foams (Supplementary Video S2). The data that support the findings of this study are available from the corresponding authors upon reasonable request.

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

## Acknowledgements
We especially acknowledge Anders Ekholm (SLU) for his valuable input in assisting with SE-HPLC measurements and Hanno Holzinger (KTH) for taking photos in the compression set test. Dr Mercedes Jiménez- Rosado (Seville University) is acknowledged for her assistance with the crosslinking tests. This work was financed by Formas (2019-00557, Mercedes Bettelli), Bo Rydins and Formas (F30/19 and 2022-00362, Antonio Capezza), and Crops for the future (CF4). Open access funding provided by Royal Institute of Technology.

## Author contributions
M.A.B. performed some samples characterizations and designed the entire study. The samples were manufactured by Q. H, who also conducted the initial sample characterizations. A.J.C carried out with [1]HNMR experiment. M.A.B. analyzed the data, made all the figures. M.A.B. write the first manuscript draft, A.J.C, E.J, R.T.O and M.S.H reviewed and revised the manuscript.

## Funding

## Competing interests
The authors declare no competing interests.
