## [Peer Review File · Communications Chemistry]

Reviewers' comments:

Reviewer #1 (Remarks to the Author):

This study added three functional additives, namely citric acid, gallic acid, and genipin, to fabricate extrudable wheat gluten foams and characterized their relative properties. It is a complete work. However, how to present this work more clearly and attractive should be taken into consideration by the authors.

1. References and citations do not match.
2. The significance and creativity of this study should be introduced in the introduction part.
3. What are the possible applications of the foams? What is the relationship between the properties characterized in the study and the possible applications?
4. Conclusion should be reorganized.
5. Some of the images have very low resolution.
6. Legends of figures and tables are missing.

Reviewer #2 (Remarks to the Author):

Review of "Effects of multi-functional additives during foam extrusion of wheat gluten materials" by Bettelli et al.

The authors in this study examined the effects of genipin, gallic acid, and citric acid on the properties of extruded wheat gluten foams. They found that the genipin samples had the most cross-linking and the citric acid samples had the lowest densities. Also, the foams had high energy loss rates under compression tests, which indicated their possible use in cushioning applications. The results are interesting and the conclusions mostly follow from the results. I recommend acceptance with minor revisions. I do have some questions:

- 1) Lines 137-139: The authors stated that "The higher efficiency of the foaming process at elevated temperatures must be weighed against the increasing aggregation of the protein at higher temperatures." Does an increase in the aggregation of gluten lead to an increase or decrease in foaming?
- 2) Lines 219-221: The authors stated that the WG/G/ABC/5GNP sample had the highest initial saline solution uptake due to the highest proportion of open pores. However, the WG/G/ABC/5GNP sample also had relatively high proportion of open pores and had low initial solution uptake. Could the authors explain this behavior?
- 3) Lines 315-317: The authors indicated that the WG/G/ABC and WG/G/ABC/1GA samples had higher β -sheet fractions than the other samples. Do the authors know why this occurred?
- 4) Lines 347-348: The authors stated "An "interaction"-induced color change is, however, normally occurring before a change can be observed in an FTIR spectrum." Could the authors explain this

statement further?

5) Lines 492-493: How was p_m determined from p_{WG} and p_G ?

6) Line 500: What is m_i in equation 5?

7) Table 1: What are the standard deviations for open, closed, and total porosity?

8) Table 6: An increase in cross-linker concentration led to a decrease in degree of cross-linking for all samples. Do the authors know why this occurred?

Comments to reviewers

Reviewer 1

Comment: References and citations do not match.

Response:

The references have been adjusted.

Comment: The significance and creativity of this study should be introduced in the introduction part.

Response:

Many thanks for this point, we have clarified the significance and creativity of using inexpensive and non-toxic multifunctional additives to potentially improve the mechanical properties (sealing/cushioning), as well as, absorption properties. Also to be able to tailor the foam structure.

Comment: What are the possible applications of the foams? What is the relationship between the properties characterized in the study and the possible applications?

Response:

The high energy loss rate under cyclic loading and the low compression set indicated that the potential applications of the foams are as cushioning materials, where the impact is relatively slow (such as in training mats), and as a sealing material in e.g. window frames. The rapid and relatively high uptake (depending on the recipe) of saline (body fluid simulant) in the swelling capacity tests indicated that the foam may be used as an absorbent material in e.g. women's sanitary products, where a medium-high liquid uptake is required (30-40 g/g per period).

Comment: The conclusion should be reorganized.

Response:

We have now reorganized the conclusions to s made them clearer.

Comment: Some of the images have very low resolution.

Response:

This has now been corrected

Comment: Legends of figures and tables are missing.

Response:

This has now been corrected

Reviewer 2:

Comment: Lines 137-139: The authors stated that “The higher efficiency of the foaming process at elevated temperatures must be weighed against the increasing aggregation of the protein at higher temperatures.” Does an increase in the aggregation of gluten lead to an increase or decrease in foaming?

Response:

As we have seen, with the use of SBC in the previous article (reference 18), a large protein aggregation occurred, which also involves intermolecular disulphide crosslinking, thus

decreasing foamability. We have rewritten the sentence to clarify that aggregation also involves crosslinking.

Comment: Lines 219-221: The authors stated that the WG/G/ABC/5GNP sample had the highest initial saline solution uptake due to the highest proportion of open pores. However, the WG/G/ABC/5GNP sample also had relatively high proportion of open pores and had low initial solution uptake. Could the authors explain this behavior?

Response:

We assume here that the reviewer means WG/G/ABC/1GNP. In this case, this material (1wt.% GNP) presents higher density, less porous structure with a higher proportion of closed pores compared to the 5wt.% GNP. These aspects considerably affect saline absorption, especially in the initial uptake that relies on capillary action.

Comment: Lines 315-317: The authors indicated that the WG/G/ABC and WG/G/ABC/1GA samples had higher β -sheet fractions than the other samples. Do the authors know why this occurred?

Response:

We thank the reviewer for this comment. Yes, this was also a bit puzzling for us, even though the spectra did not show large β -sheet formation (the spectra were relatively flat and not peaking towards lower wavelength typical of high β -sheet content). To ensure the accuracy of the data, we remeasured all samples and now all curves show the expected spectral shape indicative of low aggregation, expected at this low processing temperature. There may be some slight local fluctuation in structure that may explain the difference between the previous and new measurements. However, we prefer not to speculate about it but encourage further studies with a focus on changes in protein secondary structure and physical-chemical changes.

Comment: Lines 347-348: The authors stated “An “interaction”-induced colour change is, however, normally occurring before a change can be observed in an FTIR spectrum.” Could the authors explain this statement further?

Response:

Many thanks for this point. This is the personal experience of the authors. The initial yellowing of an aged plastic often does not show up in the FTIR spectrum. However, we have looked for references backing up this statement, and this has made us rephrase our hypothesis. The reason for that the color change did not show up in the FTIR spectrum could be due to limitations in the FTIR resolution and/or that the color change did not involve the FTIR spectral range. We have rewritten that part and also added two references;

<https://dor.org/10.1039/b511243e>

<https://doi.org/10.1520/JFS16094J>

Comment: Lines 492-493: How was ρ_m determined from ρ_{WG} and ρ_G ?

Response:

It was determined using the composite equation now included in the manuscript (equation number 4 in the revised manuscript).

Comment: Line 500: What is m_i in equation 6?

Response:

Thanks for pointing out that we missed mentioning this. It is the mass of the sample immersed in the liquid. We have clarified this now.

Comment: Table 1: What are the standard deviations for open, closed, and total porosity?

Response:

Thanks for this comment. We have now inserted the standard deviations in Table 1.

Comment: Table 6: An increase in cross-linker concentration led to a decrease in degree of cross-linking for all samples. Do the authors know why this occurred?

Response:

Thanks for this comment. Actually, this is probably a coincidence since all values are insignificantly different, except for the gallic acid 5 wt%, which showed a clearly lower degree of crosslinking, in line with also HPLC data. After careful examination of the samples we also decided to remove the genipin values from the table, since the blue color associated with the genipin reactions could interfere with the detection system of this method.

REVIEWERS' COMMENTS:

Reviewer #2 (Remarks to the Author):

The authors have satisfactorily addressed all the questions and comments.